# Converting histidine-induced 3D protein arrays in crystals into their 3D analogues in solution by metal coordination cross-linking

Xiaoyi Tan[1], Hai Chen[1], Chunkai Gu[1], Jiachen Zang[1], Tuo Zhang[1], Hongfei Wang[2] & Guanghua Zhao [1✉]

Histidine (His) residues represent versatile motifs for designing protein-protein interactions because the protonation state of the imidazole group of His is the only moiety in protein to be significantly pH dependent under physiological conditions. Here we show that, by the designed His motifs nearby the $C_4$ axes, ferritin nanocages arrange in crystals with a simple cubic stacking pattern. The X-ray crystal structures obtained at pH 4.0, 7.0, and 9.0 in conjunction with thermostability analyses reveal the strength of the $\pi$–$\pi$ interactions between two adjacent protein nanocages can be fine-tuned by pH. By using the crystal structural information as a guide, we constructed 3D protein frameworks in solution by a combination of the relatively weak His–His interaction and $Ni^{2+}$-participated metal coordination with Glu residues from two adjacent protein nanocages. These findings open up a new way of organizing protein building blocks into 3D protein crystalline frameworks.

[1] College of Food Science and Nutritional Engineering, China Agricultural University, Key Laboratory of Functional Dairy, Ministry of Education, 100083 Beijing, China. [2] Key Laboratory of Chemical Biology and Molecular Engineering of Education Ministry, Institute of Molecular Science, Shanxi University, 030006 Taiyuan, China. ✉email: gzhao@cau.edu.cn

Molecular self-assembly is essential for all living organisms, from the simplest to humans. As the most sophisticated supramolecular materials in nature, proteins, through molecular self-assembly, were utilized as standard chassis for constructing a large variety of periodic protein arrays which constitute a major component of the cellular machinery[1–3]. The ability of protein to recognize and bind to other proteins is an important factor to perform function. There is no doubt that protein–protein interactions (PPIs) are the most basic activity in most cellular functions. Protein assembly in nature is mainly directed by PPIs, which is mainly mediated by noncovalent interactions and protein geometric pattern. These arrays in cells provide mechanical support, determine cell shape, and allow movement of the cell surface to perform myriad functions[4,5]. In this respect, the exquisite, error-free self-assembly of protein macromolecules into precise organization[6–12] is a hallmark of life. Such naturally occurring protein assemblies have been extensively studied by synthetic biologists and chemists in the field of nanoscience for exploring biomimetic methods to prepare new nanoscale protein materials with valuable functions[13–20]. Importantly, some considerable applications of designing protein and constructing multifarious supramolecular protein material by varied strategies[11,21–32] for scientists had been carried out to rival the size and functionality of natural protein. These engineered protein assemblies are competitive owing to their excellent potential biocompatibility and biofunctionality. Despite these advances, there are still many anticipating and unapplied assembly strategies worthy of exploring.

The characteristics and features of constructed protein assemblies in dynamic behavior, stimuli-responsive properties, assembly efficiency and specific functions are largely associated with the intermolecular driving force involved in encoding the precise organization of protein blocks. By now, multiple interactions such as metal coordination bonds[23], disulfide bond[22], hydrogen bonds[33], electrostatic attractions[34], and hydrophobic interactions[35,36] have been extensively applied to dedicate the assembly of protein blocks into ordered architectures with distinct properties and functions. π–π stacking interactions coming from naturally occurring aromatic amino acid residues including phenylalanine (Phe), tyrosine (Tyr), tryptophan (Trp), and histidine (His) likewise play an important role in self-assembly of protein architectures. However, π–π stacking configuration from these amino acid residues are actually complicated, which consist of at least three different types of geometries that differ by the angle between rings and offset values: edge-to-face (T-shaped), face-to-face, and parallel displaced (offset stacked) interactions[37]. Our recent studies showed that 2D protein nanocage assembly can be achieved by Phe-Phe interactions at protein interfaces, while π–π stacking interactions coming from Tyr residues enable the same protein building blocks to assemble into 3D protein superlattices; unfortunately, because of lack of the crystal structure, the detailed mechanism of the above Phe-induced or Tyr-induced π–π stacking interactions remains unknown[38]. Among naturally occurring four aromatic amino acids, only His residue, with p$K_a$ range of 6.0–7.0, ionizes within the physiological pH range, and thus its p$K_a$ permits the side chain imidazolyl of histidine residue to occur as the neutral form or the protonated form[39–41]. Such valuable property raises the possibility that π–π stacking interactions from His-His pair in protein could be controlled by pH, one of external stimuli. However, to date, using π–π stacking from His–His interactions to achieve high-ordered protein arrays has yet to be explored.

We are interested in the construction of well-organized protein arrays involved by intermolecular aromatic interactions such as His–His interactions because they are reversible, chemically tunable, easily designed and engineered. We believe that highly ordered protein self-assemblies could be constructed by a combination of the aromatic interactions from His residues and protein symmetry. Herein, by single His mutation on ferritin outer surface close to its $C_4$ symmetry axes as shown in Fig. 1, we implemented the His–His interactions within two neighboring protein molecules could be fine-tuned by pH in crystals (Fig. 1b). By incorporation of metal coordination between two adjacent protein molecules as guided by the crystal structure of His mutation ferritin, we are able to construct 3D porous protein nanocage arrays in solution (Fig. 1c).

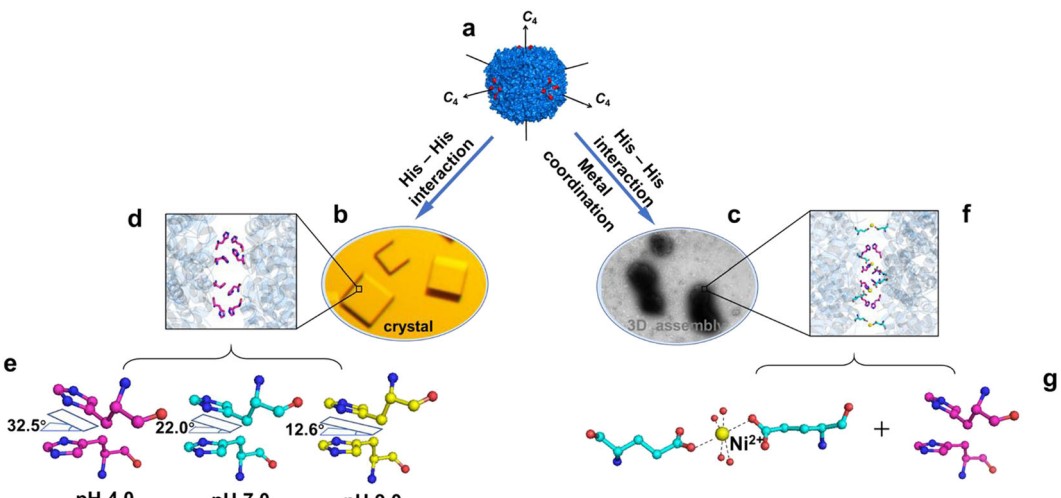

**Fig. 1 Proposed model of His-induced 3D protein crystals and 3D solid arrays in solution by combining His-His interactions and metal coordination.** **a** Close-up view from three $C_4$ rotation axes of ferritin which are perpendicular to each other. Single His mutation of each ferritin subunit on the protein outer surface nearby the $C_4$ rotation axes were highlighted in red. **b** Crystal diagram of ferritin induced by His–His interactions along the $C_4$ rotation axes. **c** Solid 3D assemblies produced by a combination of His-His interactions and metal coordination. **d** Crystal structure illustration of four pairs of His–His interactions (colored magenta) between two adjacent ferritin molecules. **e** The action mode of His–His interactions can be adjusted by pH. **f** Crystal structure revealed that both His-His (colored magenta) interaction and metal coordination of $Ni^{2+}$ (colored yellow) with glutamic acid (colored cyan) are responsible for the formation 3D protein arrays. **g** Close-up views of $Ni^{2+}$ induced metal coordination and His-His interaction in **f**.

## Results

**Design and characterization of protein nanocages as building blocks.** Recently, protein nanocages have received considerable attention due to their various functions[8,18,26,42,43]. As a standard structural component among these protein nanocages, ubiquitous ferritin is usually composed of 24 similar subunits assembling into a quasi-spherical shell with octahedral symmetry, which contains three $C_4$, four $C_3$, and six $C_2$ rotation axes[44]. Three key properties appear to make the locations near $C_4$ rotation axes as optimal sites for anchoring histidine: (1) Three $C_4$ rotation axes are perpendicular to each other, which are similar to the space coordinates X, Y, and Z as shown in Fig. 1a; therefore, high-ordered protein arrays would be formed if enough driving forces were provided along the $C_4$ rotation axes; (2) A large body of works have shown that the channels along the $C_3$ axes are major pathways for iron entrance into the ferroxidase site and protein cavity[45,46], so one could damage such ferroxidase function if the protein surface close to the $C_3$ axes was engineered; (3) Similarly, the protein surface nearby the $C_2$ axes has been reported to be involved in binding of H-type ferritin to its receptor TfR1[47].

In this work, we chose recombinant *Marsupenaeus japonicus* ferritin (MjFer) as building blocks to construct protein assemblies due to its easy purification and high yield, and focused on His-His interactions between protein nanocage molecules. We wondered whether high-ordered nanocage arrays would be created through $\pi-\pi$ interactions after a certain amino acid residue located on the protein exterior surface nearby the $C_4$ rotation axes was replaced with His residue. Analyses of the crystal structure of MjFer (PDB: 6A4U) revealed that Thr158 is an ideal position for anchoring His motif, because the side chain of Thr158 protrudes on the outside surface of the protein cage, thereby providing sufficient response area for His to trigger protein association reaction.

To approve this idea, we made a MjFer mutant by replacing Thr158 with His, which was referred to as $^{T158H}$MjFer. After overexpression in *E. coli*, we purified it to homogeneity as suggested by native-PAGE and SDS-PAGE (Supplementary Fig. 1). Similarly, results showed that the designed $^{T158H}$MjFer nanocages stay in a monodispersed state under in 25 mM Tris-HCl, pH 8.0, just like wild-type (wt) ferritin (Supplementary Fig. 2). However, under other solution conditions (pH ranged 5.5–9.5 and NaCl concentration ranged 100 mM–700 mM), massive and regular protein assemblies were hardly observed at a wider sight by transmission electron microscopy (TEM). This is most likely because the designed His-His interactions are too weak to drive protein nanocage assemble into high-ordered arrays in solution. Consistent with this idea, the research by Rajasri et al.[48] showed that the propensity of His to interact with another His residue is only 0.76, a value much lower than other pairs of His-Phe (propensity of 1.52), His-Tyr (propensity of 1.74) and His-Trp (propensity of 1.47). Further support for this idea comes from our recent study showing that 2D ferritin assemblies could be formed by designed Phe-Phe interactions which are stronger than His–His pairs[38]. Although the designed His–His interactions hardly facilitate $^{T158H}$MjFer protein nanocages to form high-ordered protein arrays in solution, they might play a role in the formation of protein crystals.

**Structural basis of His–His interactions at different pH values.** To confirm the above view, we screened a wide range of solution conditions at 20 °C, and eventually obtained qualified cubic protein crystals at pH 7.0 for X-ray diffraction (Fig. 2a, Supplementary Table 1). Subsequently, we solved the crystal structure of $^{T158H}$MjFer at a resolution of 2.4 Å (Supplementary Table 2). As expected, the crystal structure provides unambiguous evidence for the formation of His-involved $\pi–\pi$ interactions along the $C_4$ axes

(Fig. 2b–e). Four pairs of His-His interactions are involved in the molecular contact between any two adjacent protein nanocages. Bear in mind that each ferritin molecule has six 4-fold channels, substitution of Thr158 to His along these channels produce a rigid six-way connecter; if all His residues in ferritin mutant were involved in the $\pi–\pi$ interactions, one would expect that 3D protein superlattices can be observed. Indeed, in crystals, protein nanocages are bridged together by the His–His interactions to form 3D protein frameworks with simple cubic structure (Fig. 2b). Consequently, the protein packing pattern in the crystal of $^{T158H}$MjFer is markedly different from that of wild-type ferritin molecules in crystals. Wild-type ferritin molecules in crystals usually arrange into a face-centered-cubic (fcc) structure where protein molecules usually arranged with a pattern of $C_2–C_2$ interface opposite from two adjacent molecules. In contrast, the $\pi–\pi$ attraction of His158 moves $^{T158H}$MjFer molecules from the traditional stacking patterns to $C_4–C_4$ interface opposite in the crystals, although such relative weak interaction hardly drives protein assembly in dilute solution. These findings demonstrate that the His-His interactions have a marked effect on the arrangement of ferritin molecules in crystals. Also, the arrangement of $^{T158H}$MjFer protein nanocages in crystals is also distinct from reported protein arrangement due to either the interactions between protein and dendrimer[49] or special chimeric fusion of the respective genes in the subunit[50]. Further geometry analysis for the crystal structure (Fig. 2b, c) reveals that the $\pi–\pi$ stacking interactions from two adjacent His residues have nearly face-to-face configuration.

$^{T158H}$MjFer molecules assemble into a designed packing arrangement in crystals, but remain disassociated state in solution. Such difference in assembly behavior may be due to completely different experimental conditions. For example, we found that protein cubic crystals grew under the following conditions: 2500 mM NaCl, 100 mM KH$_2$PO$_4$/Na$_2$HPO$_4$, which is completely different from solution condition. Moreover, during the process of crystal formation, much higher salt concentration was required. In addition, high supersaturation is also required to reduce the free energy barrier of pre-crystalline aggregates, thereby increasing the probability of nucleation. In our conditions, 10 mg ml$^{-1}$ protein (about 20.0 μM) were required to obtain protein crystals. We believe that such high protein and salt concentration could promote the formation of a nucleus, thereby producing protein crystals. Thus, it appears that experimental conditions have a marked effect on protein assembly.

pH is a crucial physicochemical property that influences properties of most biomolecules[51,52]. As mentioned above, protonation and deprotonation states of histidine sensitively count on pH environment due to its p$K_a$ close to neutral pH. So we envisioned that pH could have an important effect on the His–His interactions between two neighboring protein nanocages in crystals. To confirm this idea, we also grew the crystals of protein molecules at pH 4.0 and pH 9.0 (Supplementary Table 1), and finally obtained suitable single crystals for X-ray diffraction. We solved the crystal structure of protein molecules at pH 4.0 and pH 9.0, respectively, with a resolution of 1.6 Å and 2.3 Å (Supplementary Table 2). On the basis of these crystal structures, the imidazole chemistry of a pair of histidine residues is clearly different at pH 4.0, 7.0, and 9.0. Here, the pairing His–His interactions are characterized by contact distance ($R$) and orientation angle ($\theta$), where $R$ corresponds to the distance from one imidazolyl ring center to the center of the another, and $\theta$ represents the orientation angle of interplanar ring. We found that the $R$ value between the stacking imidazolyl-imidazolyl rings at these three pH values (4.0, 7.0, and 9.0) are 4.0 Å, 3.7 Å, and 3.8 Å (Fig. 2e), respectively, indicating that pH hardly affects the distance between two adjacent ferritin nanocages.

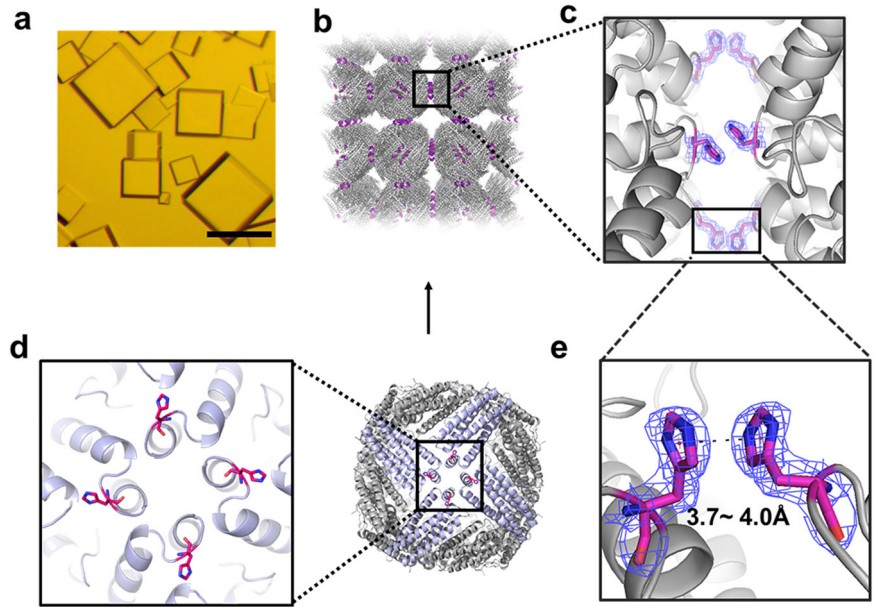

**Fig. 2 Structural basis of the His-inducible protein crystals. a** Light micrograph of cubic crystals of ^T158HMjFer. Scale bars represent 100 μm. **b** Assembly of ferritin into simple cubic (sc) packing in the crystal structure, and closeup views of the four engineered His-induced π–π stacking interactions between two adjacent protein nanocages. **c** Enlarged view of His-His chemistry between neighboring ferritin molecules. **d** The orientation of four designed His residues located on the outer surface of ^T158HMjFer nearby the $C_4$ symmetry axes, which are highlighted in red. **e** The distance between the centers of imidazole ring of His residues pairs at pH 4.0, pH 7.0, and pH 9.0 is in the range of 3.7–4.0 Å.

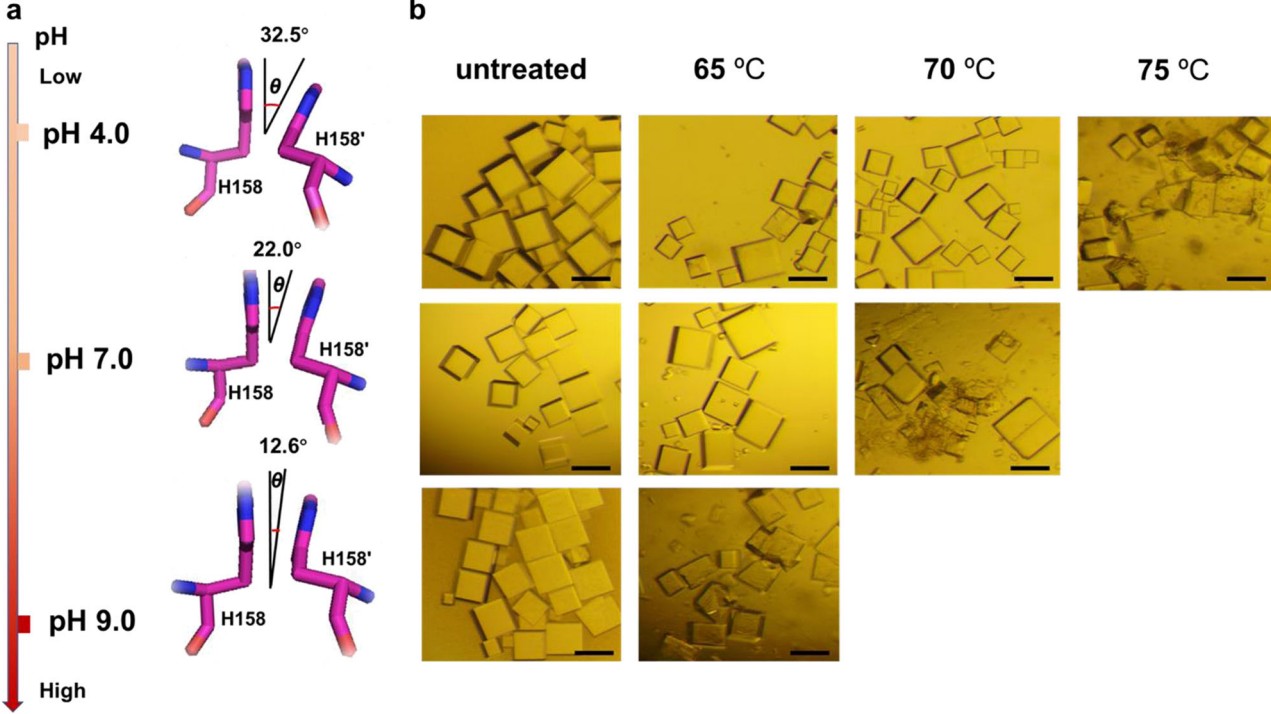

**Fig. 3 Structure difference and crystal stability difference of ^T158HMjFer crystals. a** Effect of pH on the His-induced π–π stacking interactions. The orientation angle about interplanar ring (θ) between H158 and H158′ gradually decreases as pH increases. **b** Analyses of thermal stability of protein crystals at different pH values. Scale bars represent 200 μm.

Differently, the θ value of two interactional His residues is pH-dependent based on the fact that the θ value is 32.5° at pH 4.0, 22.0° at pH 7.0, and 12.6° at pH 9.0, respectively (Fig. 3a). This is not surprising because the protonation of imidazolyl ring is highly associated with pH values. At pH 4.0, the imidazolyl ring of His is almost completely protonated, and taking net +1 charge, while it is close to a completely deprotonated form at pH 9.0. Consequently, Coulomb repulsion interactions between two protonated imidazolyl rings caused orientation angle broader, whereas two imidazolyl rings from deprotonated histidine residues are nearly paralleled due to lack of such repulsion interactions.

To elucidate which state, protonated or deprotonated, favors stronger His-His attraction, we extracted the coordination of the His-His pairs from the solved crystal structures for quantum calculations. The calculation results showed that the binding energy between His pairs at pH 4.0 ($-1.59$ kcal mol$^{-1}$) is much lower than that at pH 9.0 (45.40 kcal mol$^{-1}$), indicating that His pairs at pH 4.0 are much more stable than that of pH 9.0. Therefore, side chain interactions from protonated His pairs are stronger than that between their neutral or deprotonated analogues, being in agreement with the previous statistical and computational study[53].

To confirm the above conclusion, we further determined the thermostability of the crystals grown at different pH values (4.0, 7.0, and 9.0) with thermal treatment at different temperatures (60, 65, 70, and 75 °C) for 15 min, respectively. As shown in Fig. 3, all crystals can be resistant toward thermal treatment at 60 °C. However, after being treated at 65 °C, some crystals obtained at pH 9.0 were damaged, while crystals at pH 7.0 and 4.0 were well-kept (Fig. 3b). When temperature increased to 70 °C, the crystals obtained at 7.0 began to be broken, whereas the crystals grown at pH 4.0 can suffer from such thermal treatment until 75 °C to show a disruptive state. Thus, it seems that the crystals at 4.0 exhibit the highest thermostability. Based on the calculation results, we speculate that the pH-dependent thermostability is associated with the increased interactions from His pairs which undergo the deprotonation gradually with pH reaching to 4.0.

## Ni$^{2+}$-directed 3D self-assembly of ferritin in solution.
$^{T158H}$MjFer molecules could not assemble into high-ordered protein superlattices in solution. This might be because the π–π interactions derived from His residues between $^{T158H}$MjFer molecules in solution are not strong enough to generate protein superlattices. To corroborate this interpretation, we tried to strengthen the interplay between two neighboring ferritin molecules through incorporating extra driving force. After analyzing the $^{T158H}$MjFer crystal structure, we found that the distance between Glu95 and Glu95′ coming from two adjacent protein nanocages, which is located at the lateral surrounding of the $C_4$ symmetry axes, is ~5.2 Å (Supplementary Fig. 3). This distance raises the possibility that a metal ion could link these two Glu residues through metal coordination, thereby facilitating the formation of 3D protein superlattices in solution. To this end, different divalent metal ions such as Ca$^{2+}$, Zn$^{2+}$, Cu$^{2+}$, Ni$^{2+}$, and Co$^{2+}$ were added to protein solution, respectively, followed by TEM analyses. Surprisingly, among those metal ions, we found that well-organized 3D protein arrays was only formed in a buffer solution (25 mM Tris-HCl, 500 mM NaCl, pH 8.0) containing 0.7 mM of Ni$^{2+}$ (Fig. 4a–c), while no such protein arrays were observed under the same experimental conditions when Ni$^{2+}$ was replaced by other metals such as Ca$^{2+}$, Zn$^{2+}$, Cu$^{2+}$, and Co$^{2+}$, respectively, suggesting that only Ni$^{2+}$ can induce assembly of monodispersed $^{T158H}$MjFer molecules into 3D protein superlattices.

It has been known that Ni$^{2+}$ ions could coordinate with multiple groups such as carboxyl group and imidazole group. To confirm whether Glu95 rather than His158 is required for the coordination with Ni$^{2+}$ ions, we made another mutant, $^{T158H/}$ $^{E95A}$MjFer, where Glu95 and Thr158 have been replaced by Ala95 and His158, respectively. If Glu95 is a key amino acid residue involved in coordination with nickel ions, one would expect that no protein array occurs with this mutant in solution under the same experimental conditions. As expected, upon adding 0.7 mM of Ni$^{2+}$ ions to the $^{T158H/E95A}$MjFer solution, no high-ordered protein assembly occurred, and instead monodispersed ferritin molecules appeared in solution (Supplementary Fig. 4a). Similar

results were obtained upon adding either 500 mM NaCl or 0.7 mM Ni$^{2+}$ plus 500 mM NaCl to the $^{T158H/E95A}$MjFer solution (Supplementary Fig. 4b, c). These findings demonstrate that Glu95 is a key amino acid residue for controlling the formation of the above 3D protein superlattices through its coordination with Ni$^{2+}$. In addition, wild-type MjFer molecules kept monodispersed constantly under all experimental conditions (Supplementary Fig. 4d–f), illustrating that the formation of the observed 3D assemblies is derived from cooperative effects of two different interactions: metal coordination and π–π interactions.

To determine whether the Ni$^{2+}$-induced assembly of $^{T158H}$MjFer molecules are structurally simple cubic lattice in solution, the above samples were further analyzed by small-angle X-ray scattering (SAXS)[26,54]. The 2D scattering pattern measured directly from a sediment of nanoparticles shows strong Bragg peaks, which indicates a high long-range order and large domain size of the nanoparticles (Fig. 4d). The diffraction maxima are found from the azimuthally integrated curve at $q = 0.052$, 0.074, 0.090, and 0.103 Å$^{-1}$ corresponding to the plane reflections of Miller indices of (100), (110), (111), and (200). The $q$ values of the sample are closely coincided with $q$: $q^* = 1$: $\sqrt{2}$: $\sqrt{3}$: $\sqrt{4}$, indicative of a cubic lattice (Fig. 4d). Moreover, the lattice constant of 11.7 nm, corresponding to the center-to-center distance between the simple cubic lattice, is in good agreement with the outer diameter of ferritin (~12 nm) (Fig. 4e).

Although SAXS analyses are able to produce crystallographic parameters (i.e., symmetry and unit cell parameters), they cannot furnish an atomic metal-mediated protein assembly information. To elucidate the detailed structure of the above 3D protein arrays, we tried to obtain qualified single crystals of $^{T158H}$MjFer in the presence of Ni$^{2+}$ based on crystallization conditions as shown in Supplementary Table 1, and subsequently solved the crystal structure at a resolution of 1.7 Å (Supplementary Table 2). As expected, intermolecular associations between ferritin molecules are indeed mediated through Ni$^{2+}$ involved metal coordination, while His158-participated π–π stacking interactions are also involved in (Fig. 5a), suggestive of their cooperative interactions. Totally, there are four metal coordination bonds and four pairs of π–π stacking interactions formed between two neighboring $^{T158H}$MjFer molecules (Fig. 5b, c), being consistent with the intended design of the side chain. The contact distance ($R$) and orientation angle ($θ$) of two interactional His residues in this Ni$^{2+}$ involved crystal structure are 3.6 Å and 28.1°, respectively (Supplementary Fig. 5), suggesting that the formation of metal coordination bond has little effect on the His-His interactions. Moreover, all nickel complexes serve as bridges in a hexacoordinated form, where equatorial plane of the quadrangular bipyramid is composed of four water molecules and two apical positions are occupied by two carboxylic groups of Glu95 and Glu95′ (Fig. 5d). A listing of final observed and calculated structure factors is available as supplementary material in Supplementary Table 2. All these findings demonstrate that a combination of π–π interaction contributed from His pairs and metal coordination from nickel ions and Glu residues from two adjacent protein nanocages can drive protein molecules assemble into high-ordered 3D arrays.

To determine whether the coordinated water molecules in the Ni$^{2+}$ bridged complexes could be replaced by other ligands, we soaked the above crystals in imidazole-containing solution (100 mM), and then solved the crystal structure at resolution of 2.4 Å (Supplementary Table 2). The crystal structure revealed that two added imidazole molecules displaced two previously coordinated water molecules, forming a distorted octahedral bipyramid geometry (Fig. 5e), and consequently two imidazole ligands and two left water ligands are not in the same plane

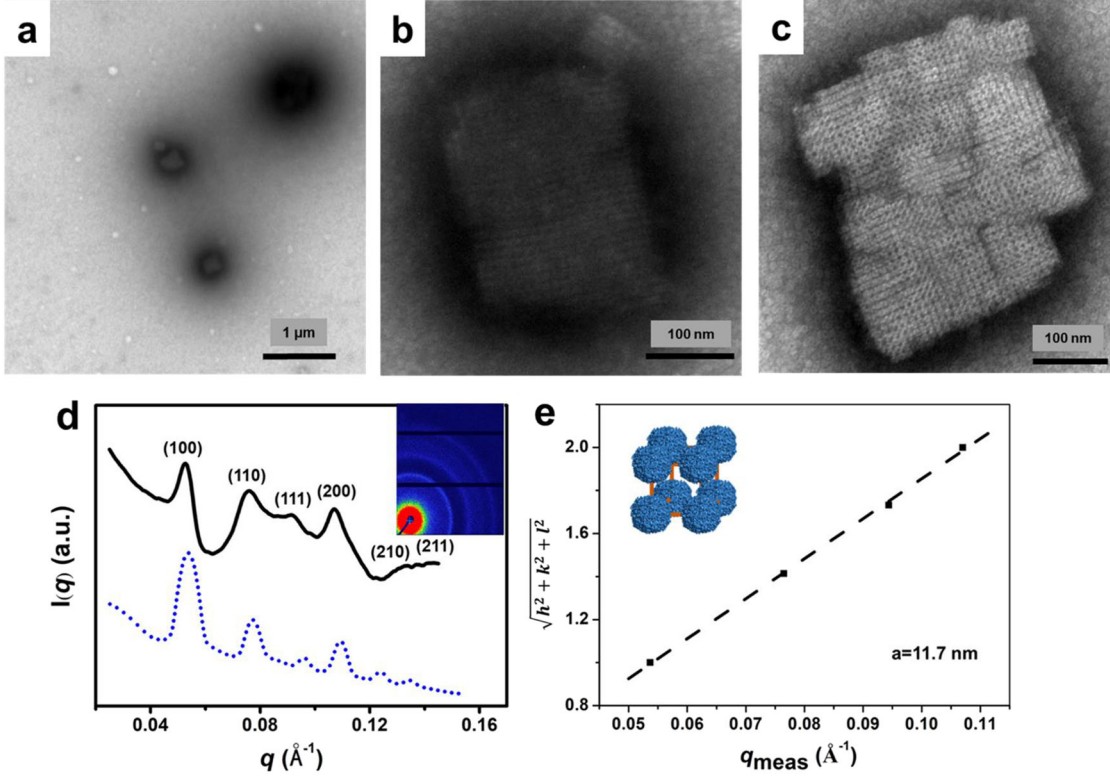

**Fig. 4 Characterization of 3D <sup>T158H</sup>MjFer superlattices.** Wait—

**Fig. 4 Characterization of 3D [T158H]MjFer superlattices. a** TEM images of 3D assembly of [T158H]MjFer formed by a combination of π–π stacking interactions and $Ni^{2+}$ coordination. **b**, **c** Enlargements of protein superlattices. **d** SAXS analyses of 3D [T158H]MjFer superlattices. The (*hkl*) values in radially averaged 1D SAXS data are labeled above the peaks. The experimental curve (black) matches the simulated pattern (blue dash) well, revealing a simple cubic (sc) structure. The inserted image in **d** is the 2D SAXS pattern of [T158H]MjFer assemblies. **e** Miller indices of assigned reflections for the sc structure versus measured *q*-vector positions for indexed peaks yield unit cell dimensions of *a* = 11.7 nm. The inserted image in **e** is a unit cell of the superlattice composition. TEM conditions: 1.0 μM of [T158H]MjFer protein in 25 mM Tris-HCl, pH 8.0 containing 500 mM NaCl, 0.7 mM of $Ni^{2+}$. SAXS samples were prepared by centrifugation of 3D assemblies.

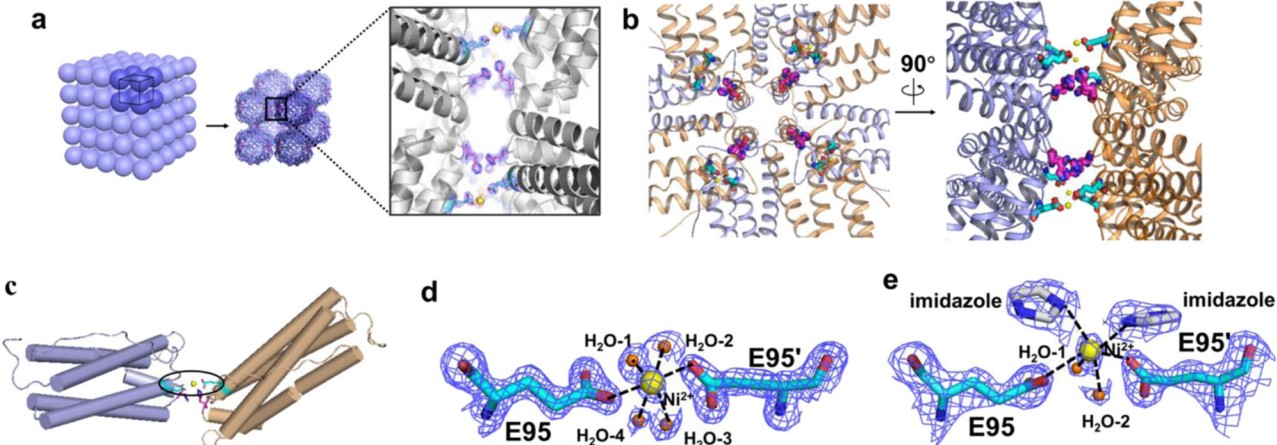

**Fig. 5 The crystal structure of 3D protein nanocage arrays. a** Assembly of [T158H]MjFer nanocages into sc packing induced by a combination of metal coordination and His-His interactions, and closeup views of π–π stacking and metal coordination. $Ni^{2+}$ ions are shown as yellow spheres, histidines and glutamic acids are shown as magenta and cyan sticks, respectively. **b** The overlook view and side view of His-participated π–π stacking interactions and $Ni^{2+}$-induced coordination between two neighboring nanocages. **c** Zoomed-in view of a couple of interactional subunits. **d** A zoomed-in view of one $Ni^{2+}$ complex where $Ni^{2+}$ is coordinated to four $H_2O$ and glutamic acid residues Glu95 and Glu95′ contributed from two protein nanocages, respectively. The $Ni^{2+}$ ions and water molecules are shown as yellow sphere and small orange spheres, respectively. The distances of $H_2O$-1, $H_2O$-2, $H_2O$-3 to $Ni^{2+}$ (2.1 Å) are identical, while the distance of $H_2O$-4 to $Ni^{2+}$ is 2.4 Å and the carboxylic group of Glu to $Ni^{2+}$ is 2.5 Å. **e** A zoomed-in view of one $Ni^{2+}$ complex where two water molecule ligands are substituted by two imidazole molecules. The distance of water molecules, imidazole molecules and the carboxylic group of Glu to $Ni^{2+}$ are 2.3 Å, 2.6 Å, and 2.9 Å, respectively. The coordination of $Ni^{2+}$ with surrounding atoms were marked with black dash lines.

(Supplementary Fig. 6a, b). Such structure is markedly different from the nickel complex formed in the absence of imidazole (Fig. 5d). Further structural analyses revealed that both the $Ni–O_{Glu}$ distance and the angle of $O_{Glu}–Ni–O_{Glu}$ were markedly changed due to replacement of $H_2O$ ligands by imidazole: 2.9 Å and 137.0° for the $^{T158H}MjFer–Ni^{2+}$-imidazole complex versus 2.5 Å and 175.0° for $^{T158H}MjFer–Ni^{2+}–H_2O$ complexes (Supplementary Fig. 6c, d). The above displacement reactions of $H_2O$ by imidazole molecules could be used to mimic the coordination environment of some nickel-containing enzymes, the catalytic center of which usually consist of N-coordinated ligands[55].

## Discussion

Side chains of amino acid residues located in the exterior surface of protein nanocages are directly involved in protein-protein interactions at the interfaces, so biological incorporation of specific amino acids into site-selective protein interfaces represents an effective method for controlling protein assembly. To this end, selecting the suitable amino acid is crucial. Among the 20 naturally α-amino acids, histidine is the most unique in that the protonation state of histidine side chain, with $pK_a$ range of 6.0–7.0, sensitively depends on pH and on the local environment of the imidazolium side chain, so the His-His interactions could be regulated by solution pH. As expected, in crystals, $^{T158H}MjFer$ molecules are packing into a simple cubic structure by the designed His-His interactions, and such packing structure is pronouncedly distinct from a face-centered cubic structure of wild-type ferritin molecules. Besides, the action mode of His-involved π–π interactions is pH-dependent as revealed by the crystal structure at a high-resolution at different pH values. Consequently, the His-involved π–π attraction increases as pH decreases from 9.0, 7.0, to 4.0. This represents the first report that the π–π interactions from His-His revealed by crystal structure could be designed to be controlled as a function of pH.

While a variety of different dimensional protein assemblies have been reported by using different noncovalent and covalent interactions as driving forces[11,21–32], the majority of the designed PPIs for high ordered protein assemblies is based on single noncovalent or covalent interaction as a driving force. Here, we demonstrate that a combination of π–π interaction and metal coordination represents an effective way to build 3D protein arrays in crystals and solution. The structure of the resulting 3D protein nanocage frameworks was characterized by X-ray protein crystallography at an atomic level, which emphasizes the importance of cooperation of multiple chemical interactions. As compared to the reported approach for the creation of protein assemblies that needs intensive re-engineering of protein interfaces, the reported strategy here that focuses on single mutation in conjunction with the symmetry of protein building blocks is conceptually and operationally simple, which are certainly with great promise to the construction of other protein arrays with solid assemblies.

## Methods

**Cloning, overexpression and purification of proteins.** The mutagenesis of $^{T158H}MjFer$ was performed by a fast site-directed mutagenesis kit (TIANGEN Biotech Co., Ltd.), after the MjFer (*Marsupenaeus japonicus* ferritin) plasmid was cloned into the pET-3a vector (Novagen). DNA fragments were amplified (18 cycles with denaturation for 20 s at 94 °C, annealing for 30 s at 55 °C, and elongation for 2.5 min at 68 °C. Subsequently, a final cycle extension for 5 min at 68 °C using the primer pairs was performed, and then *Dpn* I enzyme was added to the DNA template to incubation for 1 h.

Wild-type ferritin was expressed and purified as reported previously[42]. Mutant plasmids were introduced into *E. coli* strain BL21 (DE3) and plated on LB agar containing ampicillin. 500 mL LB-Miller medium with 50 μg ml$^{-1}$ ampicillin were inoculated with the preculture and incubated at 37 °C and 200 rpm. Once the $OD_{600}$ reached ∼ 0.6, 1.0 mM isopropyl-β-D-1thiogalactopyranoside (IPTG) was added to induce protein overexpression. Cells were harvested by centrifugation at 10,000 rpm for 10 min and suspended in a buffered solution containing 25 mM Tris (pH 8.0), followed a disruption by sonication. Subsequently, the supernatant collected was subjected to precipitation with 20 % ammonium sulfate. The pellet was re-dissolved and dialyzed against the buffer of 25 mM Tris (pH 8.0). Resulting crude protein solution was subjected on ion-exchange column (Q-Sepharose Fast Flow, GE Healthcare) with a gradient elution from 0 to 1000 mM NaCl. After protein purity was analyzed via SDS-PAGE (polyacrylamide gel electrophoresis), the resultant protein was collected for the following experiments.

**Transmission electron microscopy (TEM) analysis.** Usually 10 μl droplet of samples were placed on carbon-coated copper grids and the excess sample was blotted away with filter paper. Then samples were negatively stained with 2% uranyl acetate. TEM micrographs were obtained using a Hitachi H-7650 transmission electron microscope at 80 kV.

**Protein crystallization, data collection and data processing.** Purified $^{T158H}MjFer$ was concentrated to about 10 mg ml$^{-1}$ in 10 mM Tris-HCl (pH 8.0) buffer. 1.50 μl aliquots of the protein sample was mixed with an equal volume of mother solution and the mixture was equilibrated against 500 μl mother solution at 20 °C. 24-well plates for hanging-drop vapor diffusion were set up. The crystallization conditions are showed in Supplementary Table 1. Hexahedron-shaped crystals appeared in 1day and matured within 1 week. The crystals were cryo-protected in a solution containing 25% glycerol and transported to synchrotron-radiation source. Diffraction data were collected at the beamline BL17U and BL18U at Shanghai Synchrotron Radiation Facility (SSRF) with indexing, integrating and scaling by HKL-3000 software. The structures of all protein samples were solved by molecular replacement using the wild type *Marsupenaeus japonicus* ferritin as initial model (PDB code: 6A4U) using the MOLREP program in the CCP4 program package. Refinement was conducted with the program REFMAC5 and structure rebuilding with the program COOT. The program PyMOL was employed for figure preparation. Data collection statistics and refinement details are shown in Supplementary Table 2.

**Small angle X-ray scattering (SAXS) analyses.** Small angle X-ray scattering measurements were experimented using a Xeuss2.0 SAXS/ WAXS system (Xenocs, France). Blank sample with no protein was collected before experiment sample and a standard sample (silver behenate) were used to calibrate the scattering length vector $q$. Experimental sample was obtained from sediment after centrifuging when 3D assemblies formed. The magnitude of the scattering vector $q$ is given by $q = 4 \pi \sin \theta/\lambda$, where $2\theta$ is the scattering angle. Sample thickness was approximately 1.5 mm. The simulated scattering patterns were obtained with Scatter (version 2.5).

**Computational experiment.** In order to investigate the binding energy of His pairs ($\Delta E$) between two adjacent ferritin nanocages, quantum chemical calculations were carried out. $\Delta E$ is expressed as the difference in electronic energies of the complex and individual His residue. $\Delta E = E_c - E_{m1} - E_{m2}$, where $E_c$ is the potential energy of the complex of His pairs, $E_{m1}$ and $E_{m2}$ is the energy of individual His residues, respectively. The coordinates of all possible His pairs were extracted for the quantum chemical calculations. The calculation was carried out in the solvent model density (SMD) water model[56], using density functional theory (DFT), M062X functional[57], and ma-def2-TZVP basis set[58] for single point energy calculation, respectively. All the calculation was carried out with Gaussian09.

## Data availability

The coordinates for the X-ray structures have been deposited to the Protein Data Bank (PDB) with accession PDB IDs: 6LS2 for $^{T158H}MjFer$ at pH 4.0), 6LRW for $^{T158H}MjFer$ at pH 7.0), 6LRV for $^{T158H}MjFer$ at pH 9.0), 6LRX for $^{T158H}MjFer–Ni^{2+}–H_2O$ complex, and 6LRU for $^{T158H}MjFer–Ni^{2+}$-imidazole complex, respectively. Other data are available from the corresponding author upon reasonable request.

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

## Acknowledgements

This work was supported by the National Natural Science Foundation of China (Nos. 31730069 and 31972018). The Shanghai Synchrotron Radiation Facility (SSRF) is especially acknowledged for beam time from BL17U1/BL18U1beamline.

## Author contributions

G.Z. conceived and directed the project and wrote the paper. X.T. performed experiments, analyzed data, and co-wrote the paper. H.C. performed the crystal data collection and solved the crystal structures. C.G performed crystallization experiments. J.Z., T.Z., and H.W. solved the crystal structures and co-wrote the paper.

## Competing interests

The authors declare no competing interests.
