## [Peer Review File · Communications Chemistry]

Reviewers' comments:

Reviewer #1 (Remarks to the Author):

In this manuscript, the authors report a metal-mediated approach to prepare 3D crystalline assemblies of protein nanocages. Improved upon the previous work by Zheng et al (ref 21), which focused on 2D crystalline assemblies, this study introduced histidine mutations and metal-binding configurations to synthesize 3D crystals via increased inter-cage interactions. Depending on assembly conditions, 2D crystals were generated by pi-pi stacking between histidine mutations, whereas 3D crystals of protein nanocages were only formed with the presence of Ni²⁺. Overall, this work is interesting and well-characterized. I would recommend its publication after the following comments/questions are properly addressed:

1. As stated in the title "metal coordination cross-linking", the authors classify the interaction between cages to be metal coordination bond, however, this argument is controversial. According to Fig. 6d-e, the two types of Ni-O (Glu) bond distance are 2.5 and 2.9 Å, respectively. Typically, a Ni-O coordination bond is ~ 2 Å, and the bond distances observed in this study are not only much longer than that but also longer than the Ni-OH₂ bond (2.4 and 2.3 Å). Such observation suggests the interaction could be electrostatic rather than metal coordination. Although the authors tested a series of metal ions in their study, the role of Ni during crystallization is still somewhat unclear. Since the metal-cage interaction is central to this manuscript, it is important for the authors to discuss the metal-mediated bonding interactions with further details, and revise the manuscript accordingly (title and abstract).

2. It would be important for the authors to include the pH conditions for obtaining 2D assemblies, such that the crystallography-based structural analyses can be better understood by its correspondence.

3. Line 206-209, the authors claim that pi-pi interactions from His pairs can only facilitate the formation of 2D protein arrays. However, in section "Structural basis of protein nanocage arrays induced by His-His interactions at different pH conditions", the authors already obtained many 3D single crystals from only His pair interactions. This part is self-contradictory as crystallization itself is a 3D assembly in solution. If micron-sized single crystals can be obtained, it is very likely to get much smaller ones as in Fig. 5 by some variations in assembly condition. It appears to the reviewer that decreasing pH can also make strong enough interaction for 3D assemblies. In the revised manuscript, the authors are expected to resolve this contradiction between the experimental results and the conclusion.

4. In the supporting information, Table S1, there are typos in buffer composition "KH₂PO₃/KH₂PO₃".

5. In Figure S7, high sample concentration prevents convincing assignment of sample morphologies - e.g. whether they are 2D or 3D assemblies. I would suggest imaging the sample at a lower concentration or performing a tilt-series experiment.

Reviewer #2 (Remarks to the Author):

This manuscript describes the use of His residues as chemically switchable handles to control the assembly state of ferritin. Briefly, the authors have incorporated His residues near the four-fold symmetry axes of ferritin to enable the protein to self-assemble. They observe that the protein indeed forms two-dimensional lattices upon increased ionic strength which promotes pi-pi interactions between the His sidechains. They were able to structurally characterize the His-His pi-stacking interactions by obtaining three-dimensional crystals at very high salt concentrations. The

thermostability of these crystals were found to be different at different pH's (high stability at high pH; low stability at low pH), which the researchers have attributed to variations in the strength of His-His pi-stacking interactions. The authors further observed that the formation of three-dimensional ferritin crystals could be further promoted by metal coordination to glutamate residues across the same ferritin-ferritin interface where the His-His interactions are present. Overall, these are new observations based on solid structural data that are certainly publishable eventually after additional experiments and major re-writing (see below). More importantly, however, I believe that the findings of this study are not particularly novel in relation to extensive work on protein assembly in the last decade and particularly in the case of ferritin (see below). Additionally, the scope of the study is not sufficiently broad to make claims about generalizability or creating a new approach as the authors claim.

1) Regarding novelty, the statement in the introduction "Despite these advances, rendering conversion of protein supramolecular protein assemblies with different dimensions controllable by design remains challenging." is simply not true. There are so many examples where such control has been achieved in the last 7-8 years that even references 21-32 (which by themselves invalidate the statement) cover only a tiny subset of references that should be cited and discussed in further detail. Along these lines, the authors then make a second statement "However, to date, the design of the 2D or 3D protein arrays by using π - π stacking interactions from His residues has largely been inaccessible." to motivate their study, but this then becomes such a narrow premise that at this level of "specificity" anything would become novel. This is exacerbated by the fact that there has been a lot of work on ferritin self-assembly in the recent years, whereby the self-assembly of this protein into various two- and three-dimensional arrays have been mediated by electrostatic, metal, disulfide and pi-stacking interactions, taking advantage of the high symmetry of this protein – all of these works on ferritin should have been discussed more explicitly, as they are so relevant to the current work. In fact, the current work can essentially be considered as a direct follow-up to and therefore an incremental advance over the recent studies by G. Zhao et al., for example: ACS Nano 2018 "On-Axis Alignment of Protein Nanocage Assemblies from 2D to 3D through the Aromatic Stacking Interactions of Amino Acid Residues", ChemComm 2019 "Self-assembly of engineered protein nanocages into reversible ordered 3D superlattices mediated by zinc ions", Nano Lett, 2019 "Designed Two- and Three-Dimensional Protein Nanocage Networks Driven by Hydrophobic Interactions Contributed by Amyloidogenic Motifs" etc.). The fact that these highly relevant studies—which are all from the same group—are not mentioned or discussed in the introduction creates the impression of an attempt at inflating the novelty of the current work, significantly lowering the scholarship. Along these lines, the authors adapt many phrases from other papers in the literature without explicitly citing them or mentioning them. There is absolutely nothing wrong in emulating or paraphrasing, especially if this is used to make an important point. What is less than desirable is not doing a proper job covering and citing relevant literature.

2) Thus, in light of the fact that other aromatic interactions have been used previously to drive 2- and 3-assembly of ferritin in an ionic-strength dependent fashion, the His-His interactions employed here are not novel per se. What is potentially more interesting is the potential of fine-tuning other properties by pH. In this regard, the pH-dependence of the 3D crystal thermostability is interesting. However, the authors make a big assumption in tying the thermostability of ferritin crystals solely to His-His interactions. While this fits the narrative of the paper, it is far from being definitive, as changing the pH from 4 to 9 not only alters the His protonation state of His residues but also the overall charge on the ferritin molecules and the protonation state of other interfacial residues (including Glu, Asp and Lys residues), among others. The His-His angle data from crystal structures is interesting but not definitive. Furthermore, the authors cite a J. Phys. Chem. paper (reference 43) in support of attractive interactions between protonated His-His sidechains. The interactions studied in that paper occur at quite different geometries than what is observed in the crystal structures. In a future submission, the authors may wish to do calculation of His-His energies in different protonation states in the observed geometries.

3) The metal-coordination interactions mediated by Glu residues appear to be an afterthought rather than design. In fact, a reasonable first expectation would have been that the His residues should coordinate Ni and Cu before the Glu residues did (the authors really did not consider this initially?). Also, it appears that ionic strength itself is strong enough to drive 3D assembly. A good control experiment would be to eliminate the Glu's and examine the self-assembly in that case. Regardless of the author's original intention, using His's for pi-stacking interactions and nearby Glu's for metal coordination for controlling protein self-assembly is not likely to be a generalizable strategy. On a related note, the concluding statement "This new approach represents a straightforward manner and emphasizes the importance of a combination of multiple chemical interactions in constructing 3D protein arrays." Is again not accurate. Again, there are too many studies in the literature that have exploited multiple types of interactions to control 3D protein self-assembly. From their prose and selection of papers to cite, the authors appear to be aware of such studies and I encourage them to be more thorough and accurate in the future.

4) The results on imidazole binding to Ni centers do not add anything to the study. Regardless, the authors should show the electron densities of all ligands).

5) I believe that the following statement in lines 117-120 has a faulty logic: "Such difference between protein assembly in solution and in crystals suggests that the strength of the His-involved π - π interactions in solution is weaker than that in crystals. If strong driving forces were provided along the C4 axes of ferritin, 3D protein assembly would be likewise generated in solution". All of the interactions (His-His or Glu metal) are equally available to mediate self-assembly in 2 or 3D, as ferritin has 3D symmetry. In fact, one could argue that if the His-His interactions were weaker in solution, then they would tend to favor 3D assembly as this would increase the multiplicity of these interactions. This may actually indicate that a) the ferritin-ferritin interaction geometry is different in 2D and 3D lattices and/or b) there are other factors besides His-His interactions that direct self-assembly pathways. The authors may want to rethink their logic.

Reviewer #3 (Remarks to the Author):

The manuscript entitled "Converting histidine-induced 2D protein arrays into their 3D analogues in solution by metal coordination cross-linking" describes an interesting study to control and tune the assembly behaviour of ferritin protein cages. Genetic modifications of proteins to create various lattice structures has gained increasing attention (A. Tezcan group, Nature) and hence I find the manuscript topical and timely. I particularly appreciate the extensive crystallographic analysis at different pH values. The results are communicated clearly, although the inset in Figure 5 seem to be distorted. The manuscript could become publishable after the following major revisions.

1. Abstract and line 121 states that: histidine residues have a unique pKa. Why is the pKa unique? This statement is not accurate.

2. Abstract: "strength of the π - π interactions between two adjacent protein nanocages can be fine-tuned by pH". Why are the π - π interactions tuned and not for example electrostatic? I would assume that the effect of pH is far greater in the case of electrostatic interactions. How can the authors rule out the role of other interactions? This should be discussed in detail.

3. 2D arrays in Figure 2b, should be verified with cryo-TEM or solution SAXS. The assembly could easily be a surface drying effect.

4. Why was the buffer NaCl concentration lower in pH 4 crystallization experiments?

5. Tuning the structure of ferritin crystals is highly relevant for this study: publications such as

ACS Nano, 2015, 9, 11278-11285 and Biomacromolecules, 2015, 16, 2006-2011 should be cited.

6. Experimental details given for TEM, SAXS, DLS etc. do not allow reproduction of the experiments. E.g. what were the solution conditions and concentration of ferritin? Experiments must be described in detail.

7. Line: 80 spelling mistake "sufficient".

8. Line: 86 spelling mistake "buffer". This seems to be consistently wrong in the manuscript. Please check all double f.

9. Line 146, comma typo.

10. Line 148: the statement about crystals stability at pH 4 seems to be a bit mixed up. Please revise the text.

11. Line 171-172: unclear sentence.

12. First sentence of "Discussion" is very unclear. Please revise.

13. Line 438: spelling mistake "buffer".

Response to comments from reviewer 1:

1) As stated in the title “metal coordination cross-linking”, the authors classify the interaction between cages to be metal coordination bond, however, this argument is controversial. According to Fig. 6d-e, the two types of Ni-O (Glu) bond distance are 2.5 and 2.9 Å, respectively. Typically, a Ni-O coordination bond is ~ 2 Å, and the bond distances observed in this study are not only much longer than that but also longer than the Ni-OH₂ bond (2.4 and 2.3 Å). Such observation suggests the interaction could be electrostatic rather than metal coordination. Although the authors tested a series of metal ions in their study, the role of Ni during crystallization is still somewhat unclear. Since the metal-cage interaction is central to this manuscript, it is important for the authors to discuss the metal-mediated bonding interactions with further details, and revise the manuscript accordingly (title and abstract).

Response: Thanks for the detailed coordination interaction questions. Actually we have considered this carefully when we are doing the structure Refining. Indeed, the optimized Ni-O coordination bond in metalloprotein is around 2.3Å. When we use the default Ready Set program in Phenix to optimize the metal coordination in our protein assembly structure, we got the Ni-O distance at 2.33 Å, which seems to be optimized to the average Ni-O distance, however, the Glu95 did not fit the density map perfect in our high-resolution structure solution. In light of this, we release the distance restriction of Ni coordination environments, to fit the density map perfect, the final result we got is 2.85 Å (~ 2.9 Å).

Base on the data statistics from MetalPDB server (<http://metalweb.cerm.unifi.it/>), we found the distance between Ni-O in the published PDB bank could goes up to 2.894 Å (see figure show as below). considering the distance between Ni-O in the crystal structure without imidazole substitute was 2.45 Å (~ 2.5 Å, figure 6d), which was not an optimized result but definitely a reasonable Ni coordination bond, we use the metal coordination cross-linking in the title without hesitate. The distance changes in the imidazole substitute structure may come from the solvent modification from high concentration of imidazole during crystal soaking.

Additionally, our results shown in Figure 5 revealed that 3D protein nanocage assembly exhibited the highest selectivity for nickel ion among all tested divalent ions including Ni^{2+} , Cu^{2+} , Zn^{2+} , Co^{2+} , and Ca^{2+} , a finding demonstrating that metal coordination rather than electrostatic interaction is responsible for the interaction between protein nanocage and Ni^{2+} . Moreover, 0.7 mM of Ni^{2+} can induce 3D protein assembly, while 500 mM of NaCl lack such ability, again demonstrating that metal coordination of nickel ions rather than electrostatic interaction triggers protein assembly.

2) It would be important for the authors to include the pH conditions for obtaining 2D assemblies, such that the crystallography-based structural analyses can be better understood by its correspondence.

Response: Suggestion was followed. We have added the pH conditions for 2D assemblies to the revised manuscript.

3) Line 206-209, the authors claim that pi-pi interactions from His pairs can only facilitate the formation of 2D protein arrays. However, in section “Structural basis of protein nanocage arrays induced by His-His interactions at different pH conditions”, the authors already obtained many 3D single crystals from only His pair interactions. This part is self-contradictory as crystallization itself is a 3D assembly in solution. If micron-sized single crystals can be obtained, it is very likely to get much smaller ones as in Fig. 5 by some variations in assembly condition. It appears to the reviewer that decreasing pH can also make strong enough interaction for 3D assemblies. In the revised manuscript, the authors are expected to resolve this contradiction between the

experimental results and the conclusion.

Response: Thanks for your suggestion. There are some misunderstanding here. We claim that pi-pi interactions from His pairs can only facilitate the formation of 2D protein arrays in solution under low-protein and low-salt conditions: 1.0 μM of ^{158}H MjFer in 25 mM Tris-HCl, pH 8.0) containing 500 mM NaCl. However, it is hard to know detailed information about His-His interactions between two adjacent protein nanocages. To solve this problem, we tried to crystallize ^{158}H MjFer protein by optimizing crystal growth conditions and finally obtained suitable crystals by using following conditions: 2000-2500 mM NaCl, 100 mM KH_2PO_4 / Na_2HPO_4 , pH 4.0 to 9.0, which are completely different from those required for 2D protein assembly. Moreover, during crystallization, except for much higher salt concentrations, high supersaturation is also required to reduce the free energy barrier of intermolecular interactions, thereby increasing the probability of nucleation. In our conditions, 10 mg/mL protein (about 20.0 μM) were required to obtain protein crystals. We believe that such high protein and salt concentration could promote 3D protein assembly, thereby producing protein crystals. That is why we see such large difference in protein assembly between two different experimental conditions. However, the crystal structure could tell us about the detailed information about His-His aromatic interactions. We added such explanation to the revised manuscript.

4) In the supporting information, Table S1, there are typos in buffer composition “ KH_2PO_3 / KH_2PO_3 ”.

Response: Suggestion was followed and correction has been made in the revised version.

5) In Figure S7, high sample concentration prevents convincing assignment of sample morphologies - e.g. whether they are 2D or 3D assemblies. I would suggest imaging the sample at a lower concentration or performing a tilt-series experiment.

Response: Suggestion was followed. A clear new TEM image of 2D assemblies of sample at lower concentration has been used to replace the old one as Supplementary Figure 7 in the revised version.

Response to comments from reviewer 2:

1) Regarding novelty, the statement in the introduction “Despite these advances, rendering conversion of protein supramolecular protein assemblies with different dimensions controllable by design remains challenging.” is simply not true. There are so many examples where such control has been achieved in the last 7-8 years that even references 21-32 (which by themselves invalidate the statement) cover only a tiny subset of references that should be cited and discussed in further detail. Along these lines, the authors then make a second statement “However, to date, the design of the 2D or 3D protein arrays by using π - π stacking interactions from His residues has largely been inaccessible.” to motivate their study, but this then becomes such a narrow premise that at this level of “specificity” anything would become novel. This is exacerbated by the fact that there has been a lot of work on ferritin self-assembly in the recent years, whereby the self-assembly of this protein into various two- and three-dimensional arrays have been mediated by electrostatic, metal, disulfide and pi-stacking interactions, taking advantage of the high symmetry of this protein – all of these works on ferritin should have been discussed more explicitly, as they are so relevant to the current work. In fact, the current work can essentially be considered as a direct follow-up to and therefore an incremental advance over the recent studies by G. Zhao et al., for example: ACS Nano 2018 “On-Axis Alignment of Protein Nanocage Assemblies from 2D to 3D through the Aromatic Stacking Interactions of Amino Acid Residues”, ChemComm 2019 “Self-assembly of engineered protein nanocages into reversible ordered 3D superlattices mediated by zinc ions”, Nano Lett, 2019 “Designed Two- and Three-Dimensional Protein Nanocage Networks Driven by Hydrophobic Interactions Contributed by Amyloidogenic Motifs” etc.). The fact that these highly relevant studies—which are all from the same group—are not mentioned or discussed in the introduction creates the impression of an attempt at inflating the novelty of the current work, significantly lowering the scholarship. Along these lines, the authors adapt many phrases from other papers in the literature without explicitly citing them or mentioning them. There is absolutely nothing wrong in emulating or paraphrasing, especially if this is used to make an important point. What is less than desirable is not doing a proper job covering and citing relevant literature.

Response: Suggestions were followed in the revised manuscript, and we have made substantial changes in both Introduction and Discussion. We also added several literatures as references involved relevant studies to the revised version.

Although we have published several papers which are related to 2D and 3D protein self-assemblies by designing different non-covalent and covalent interactions, all of them have been lacking crystal structure, so it is difficult to provide detailed information about designed protein-protein interactions at atomic level. In this study, we reported a series of high-resolution protein structures under different conditions and thus enable us to observe pi-pi interactions at atomic level. Such information is beneficial for understanding aromatic interactions stemming from natural amino acid residues between protein molecules. So far, to the best of our knowledge, there is few reports on the atomic information about pi-pi stacking intermolecular interactions. Actually, pi-pi interactions are very complex, which include at least three different types of geometries that differ by the angle between rings and offset values: edge-to-face (T-shaped), face-to-face, and parallel displaced (offset stacked) interactions. Without help of the crystal structure of protein assembly, it would be very hard to know the mode of pi-pi stacking intermolecular interactions. Our reported crystal structure of a series of His-mediated protein assemblies presented in this work at different pH values provide direct evidence for elucidating His-mediated pi-pi stacking interactions between protein molecules at atomic level for the first time. Secondly, among these four amino acids, only His residue, with pK_a range of 6.0 ~ 7.0, ionizes within the physiological pH range, such interesting property could enable us to control His-involved pi-pi stacking interaction between protein molecules by simply controlled by pH, one of external stimuli as confirmed by the present study.

2) Thus, in light of the fact that other aromatic interactions have been used previously to drive 2- and 3-Fassembly of ferritin in an ionic-strength dependent fashion, the His-His interactions employed here are not novel per se. What is potentially more interesting is the potential of fine-tuning other properties by pH. In this regard, the pH-dependence of the 3D crystal thermostability is interesting. However, the authors make a big assumption in tying the thermostability of ferritin crystals solely to His-His interactions. While this fits the narrative of the paper, it is far from being definitive, as changing the pH from 4 to 9 not only alters the His protonation state of His residues but also the overall charge on the ferritin molecules and the protonation state of other interfacial residues (including Glu, Asp and Lys residues), among others. The His-His angle data from crystal structures is interesting but not definitive. Furthermore, the

authors cite a J. Phys. Chem. paper

(reference 43) in support of attractive interactions between protonated His-His sidechains. The interactions studied in that paper occur at quite different geometries than what is observed in the crystal structures. In a future submission, the authors may wish to do calculation of His-His energies in different protonation states in the observed geometries.

Response: Suggestion was followed in the revised manuscript. We have added the relevant energy calculation of His-His interaction to the revised version. Quantum chemical calculations about the binding energy between His pairs showed that the π - π stacking interaction energies between the His-His interactions at pH 4.0 (-1.59 kcal/mol) is much lower than that at pH 9.0 (45.40 kcal/mol). This new result approved our thermal analyses results.

Besides, crystal structure analyses revealed that protein interfaces between two adjacent protein nanocages are mainly mediated by designed histidine-histidine interactions, while electrostatic interactions from other amino acids residues between two adjacent protein molecules was hardly observed, so histidine-histidine interactions over the pH range of 4.0 to 9.0 is a major factor which are responsible for the observed different crystal thermal stabilities. We have added the above information to Discussion in the revised version.

3) The metal-coordination interactions mediated by Glu residues appear to be an afterthought rather than design. In fact, a reasonable first expectation would have been that the His residues should coordinate Ni and Cu before the Glu residues did (the authors really did not consider this initially?). Also, it appears that ionic strength itself is strong enough to drive 3D assembly. A good control experiment would be to eliminate the Glu's and examine the self-assembly in that case. Regardless of the author's original intention, using His's for pi-stacking interactions and nearby Glu's for metal coordination for controlling protein self-assembly is not likely to be a generalizable strategy. On a related note, the concluding statement "This new approach represents a straightforward manner and emphasizes the importance of a combination of multiple chemical interactions in constructing 3D protein arrays." Is again not accurate. Again, there are too many studies in the literature that have exploited multiple types of interactions to control 3D protein self-assembly. From their prose and selection of papers to cite, the authors appear to be aware of such studies and I encourage them to be more thorough and accurate in the future.

Response: Suggestion was followed in the revised version. We agree with you that the coordination of nickel ions with Glu an afterthought rather than design. We deleted “a straightforward manner” in the revised manuscript. Our idea is based on careful analyses for His-mediated protein crystal structure. Actually, there are thousands of protein crystal structure which have been solved, which represents a plentiful of valuable sources to be utilized for designing protein assembly.

4) The results on imidazole binding to Ni centers do not add anything to the study. Regardless, the authors should show the electron densities of all ligands).

Response: Suggestion was followed in the revised version, and we have showed the electron densities of all ligands. In order to determine whether the coordination water molecules can be replaced with other molecules, we chose imidazole as ligand to exchange them, and found that two water molecules have a weaker binding force and can be replaced by imidazole. Such displacement reactions could be used to mimic the coordination environment of some nickel-containing enzymes, the catalytic center of which usually consist of N-coordinated ligands.

5) I believe that the following statement in lines 117-120 has a faulty logic: “Such difference between protein assembly in solution and in crystals suggests that the strength of the His-involved π - π interactions in solution is weaker than that in crystals. If strong driving forces were provided along the C4 axes of ferritin, 3D protein assembly would be likewise generated in solution”. All of the interactions (His-His or Glu metal) are equally available to mediate self-assembly in 2 or 3D, as ferritin has 3D symmetry. In fact, one could argue that if the His-His interactions were weaker in solution, then they would tend to favor 3D assembly as this would increase the multiplicity of these interactions. This may actually indicate that a) the ferritin-ferritin interaction geometry is different in 2D and 3D lattices and/or b) there are other factors besides His-His interactions that direct self-assembly pathways. The authors may want to rethink their logic.

Response: Suggestion was followed in the revised version, and corrections have been made. We also gave an explanation of why protein assembly are so different between dilute solution and crystals.

Response to comments from reviewer 3:

1) Abstract and line 121 states that: histidine residues have a unique pKa. Why is the pKa unique?

This statement is not accurate.

Response: Suggestion was followed in the revised manuscript and we gave more explanation about pKa of histidine residues. Among the 20 natural amino acids, the protonation state of the imidazole group of histidine is the only one to be significantly pH dependent under physiological conditions, and such property makes it unique. We amended this information in manuscript and two papers as references 46 and 47 related to His residues were added to the revised version as following: Matthew, J. B. Electrostatic effects in proteins. *Annu. Rev. Biophys. Biophys. Chem.* 14, 387- 417 (1985); Martel, P. Baptista, A. Protein electrostatics. Petersen, S. *Biotechnol. Annu. Rev.* 2, 315-372 (1996).

2) Abstract: “strength of the π - π interactions between two adjacent protein nanocages can be fine-tuned by pH”. Why are the pi-pi-interactions tuned and not for example electrostatic? I would assume that the effect of pH is far greater in the case of electrostatic interactions. How can the authors rule out the role of other interactions? This should be discussed in detail.

Response: Thanks for your suggestion. Suggestion was followed in the revised manuscript and we gave more discussion and added one new calculation result to the revised version. On the one hand, the crystal analyses of ^{1158}H MjFer revealed (Supplementary Figure 4) that protein-protein interactions at the interfaces are mainly mediated by designed histidine-histidine interactions, while no other electrostatic interactions of amino acids between two adjacent molecules was observed. Besides, we have added the relevant energy calculation of His-His to the revised version. Quantum chemical calculations about the binding energy between His pairs showed that the π - π stacking interaction energies between the His-His interactions at pH 4.0 (-1.59 kcal/mol) is much lower than that at pH 9.0 (45.40 kcal/mol). On the other hand, from Figure 4, at pH 4.0, even crystal structure showed a broader angle between two imidazole groups caused by electrostatic repulsion interaction, but stronger π - π interactions made the crystals still can resist a much higher temperature treatment than that at pH 7.0 and pH 9.0. Based on these considerations, we conclude the pH changes mainly influences the strength of the π - π interactions. We have added these informations to the revised version.

3) 2D arrays in Figure 2b, should be verified with cryo-TEM or solution SAXS. The assembly could easily be a surface drying effect.

Response: To confirm the TEM observation in Figure 2b, we used dynamic light scattering (Supplementary Figure 2) to determine protein assembly in solution, which is consistent with the TEM result. This finding excluded the possibility that the TEM observation is due to a surface drying effect.

4) Why was the buffer NaCl concentration lower in pH 4 crystallization experiments?

Response: During crystallization preparation process, we performed a tilt-series experiment of NaCl concentration over the range of 1.7 M ~ 3.2 M to obtain high-quality single crystals for X-ray diffraction. Consequently, at pH 4.0, single crystals could be obtained at salt concentration (1.7 M ~ 2.7 M), but the best crystal data were collected and processed when 2.0 M of NaCl concentration was used.

5) Tuning the structure of ferritin crystals is highly relevant for this study: publications such as ACS Nano, 2015, 9, 11278-11285 and Biomacromolecules, 2015, 16, 2006-2011 should be cited.

Response: Suggestion was followed, and these two papers have been cited in the revised version.

43. Liljeström, V., Seitsonen, J., & Kostiainen, M. A. Electrostatic self-assembly of soft matter nanoparticle cocrystals with tunable lattice parameters. *ACS Nano*. **9**, 11278-11285 (2015).

44. Bellapadrona, G., Sinkar, S., Sabanay, H., Liljeström, V., Kostiainen, M., & Elbaum, M. Supramolecular assembly and coalescence of ferritin cages driven by designed protein-protein interactions. *Biomacromolecules*. **16**, 2006-2011(2015).

6) Experimental details given for TEM, SAXS, DLS etc. do not allow reproduction of the experiments. E.g. what were to solution conditions and concentration of ferritin? Experiments must be described in detail.

Response: Suggestion was followed in the revised version, and all experiments details have been added to the figure descriptions and Method section.

7) Line: 80 spelling mistake “sufficient”.

Response: Suggestion was followed, and revision has been made in the revised manuscript.

8) Line: 86 spelling mistake “buffer”. This seems to be consistently wrong in the manuscript.

Please check all double f.

Response: Thanks for your suggestion. All spell mistakes have been corrected and checked in the revised version.

9) Line 146, comma typo.

Response: Suggestion was followed, and revision has been made in the revised manuscript.

10) Line 148: the statement about crystals stability at pH 4 seems to be a bit mixed up. Please revise the text.

Response: Suggestion was followed, and revision has been made in the revised manuscript.

11) Line 171-172: unclear sentence.

Response: Suggestion was followed, and this sentence has been corrected in the revised version.

12) First sentence of “Discussion” is very unclear. Please revise.

Response: Suggestion was followed. This sentence has been corrected in the revised version.

13) Line 438: spelling mistake “buffer”.

Response: Suggestion was followed, and correction has been made in the revised manuscript.

Reviewers' comments:

Reviewer #1 (Remarks to the Author):

The authors have addressed my questions. I would recommend this manuscript for publication.

Reviewer #2 (Remarks to the Author):

I have read the rebuttal letter as well as the revised manuscript by the authors, who did additional experiments to alleviate some of the suggestions. Unfortunately, I don't feel that my most important concerns are addressed. As a general note to the authors: they put a large block of comments together and then address them en masse, and while doing so, they tend to ignore some of the reviewers' comments included in that block (see more below). They also state many times that they have considered made extensive revisions in the text, but they don't explicitly write how they actually addressed the comments and what the revised text says. This makes it extremely difficult to gauge what the authors have actually done in the revision.

1) The references 21-28 are still not an adequately broad collection of 1, 2, 3D protein assemblies obtained by design and certainly not representative of the breadth of efforts in this area: it is kind of a haphazard mixture of papers with an overrepresentation of the authors' work.

2) Regarding my previous comment 2, on the pH-dependent thermostability : the authors did some new calculations but clearly the numbers they obtain are not physically meaningful or realistic. First, they show an unfavorable energy at pH 4, meaning that His-His stacking should not mediate self-assembly, and a value of 45 kcal/mol at pH 9, which is almost the strength of a covalent bond. Second, they also dismiss the role of electrostatic interactions by stating that they don't see any direct contacts between charged amino acid side chains from neighboring ferritin molecules. This is not what was meant by my comment. Electrostatic interactions are not limited to short-range, direct contacts and particularly in a lattice of highly charged, nano-scale molecules like ferritin, the cumulative effects of electrostatic interactions in a lattice could be quite substantial (much like in a lattice of ionic compounds) and extend well over several nanometers. The authors have to take their calculations with a grain of salt and consider cumulative electrostatic effects in the lattice at different pH's.

3) In my previous comment 3, I stated: "The metal-coordination interactions mediated by Glu residues appear to be an afterthought rather than design. In fact, a reasonable first expectation would have been that the His residues should coordinate Ni and Cu before the Glu residues did (the authors really did not consider this initially?). Also, it appears that ionic strength itself is strong enough to drive 3D assembly. A good control experiment would be to eliminate the Glu's and examine the self-assembly in that case." Although the authors state that "suggestion was followed", it is not clear to me how they were followed.

First, the authors did not address my question as to whether they have really not considered that His residues would coordinate Ni and Cu before they were actually discovered to be involved in pi-pi interactions (this is particularly significant as the authors just recently published a paper on metal-mediated ferritin self-assembly. The current way the research is presented, I and many I and many readers will actually wonder on what basis the authors would have been able to predict that His-His interactions would dominate over His-Ni or His-Cu coordination. If it was a serendipitous discovery, the authors should say so and build their story from there (nothing is wrong with this). If it was not, the authors should provide some meaningful reason or discussion as to why they expected His-His stacking over His-metal coordination.

Second, I could not locate in the text whether the authors carried out the control experiments with eliminated Glu residues – if not, the authors explain why they didn't think that this was a necessary experiment. I believe this is actually critical as it relates directly to the authors' arguments regarding 2D vs 3D ferritin assemblies.

4) Regarding the last point, the authors again respond to my original comment 4 as "Suggestion

was followed in the revised version, and corrections have been made. We also gave an explanation of why protein assembly are so different between dilute solution and crystals." They again don't explain in the rebuttal letter what the explanation is. Importantly, the arguments they make about salt concentrations is more consistent with the importance of long-range electrostatic effects in governing ferritin self-assembly. Further, I don't find the explanations regarding supersaturation compelling.

Reviewer #3 (Remarks to the Author):

The authors have mostly answered my concerns. However, the reply to comment 3) is not sufficient.

"2D arrays in Figure 2b, should be verified with cryo-TEM or solution SAXS. The assembly could easily be a surface drying effect. Response: To confirm the TEM observation in Figure 2b, we used dynamic light scattering (Supplementary Figure 2) to determine protein assembly in solution, which is consistent with the TEM result. This finding excluded the possibility that the TEM observation is due to a surface drying effect."

DLS data is not able to rule out ordered structures while drying. Cryo-TEM or solution SAXS data must be provided.

Response to comments from reviewer 2:

1) The references 21-28 are still not an adequately broad collection of 1, 2, 3D protein assemblies obtained by design and certainly not representative of the breadth of efforts in this area: it is kind of a haphazard mixture of papers with an overrepresentation of the authors' work.

Response: Thanks for your suggestion, and we have revised our manuscript according to your suggestions. We added some more related literatures to the revised manuscript.

2) Regarding my previous comment 2, on the pH-dependent thermostability: the authors did some new calculations but clearly the numbers they obtain are not physically meaningful or realistic. First, they show an unfavorable energy at pH 4, meaning that His-His stacking should not mediate self-assembly, and a value of 45 kcal/mol at pH 9, which is almost the strength of a covalent bond. Second, they also dismiss the role of electrostatic interactions by stating that they don't see any direct contacts between charged amino acid side chains from neighboring ferritin molecules. This is not what was meant by my comment. Electrostatic interactions are not limited to short-range, direct contacts and particularly in a lattice of highly charged, nano-scale molecules like ferritin, the cumulative effects of electrostatic interactions in a lattice could be quite substantial (much like in a lattice of ionic compounds) and extend well over several nanometers. The authors have to take their calculations with a grain of salt and consider cumulative electrostatic effects in the lattice at different pH's.

Response: Suggestion was followed, and we have corrected our manuscript in the revised version. To clarify which states of His-His pair are more stable, we extract the His-His pair coordination at pH 4 and pH 9, and calculated the binding energy of protonated and deprotonated His-His pair. Such calculated value might be not comparable with the real energy, but their relative value is meaningful. The results showed that the binding energy between His pairs at pH 4.0 is much lower than that at pH 9.0, indicating that His pairs at pH 4.0 are much more stable than that of pH 9.0. Thus, interactions between protonated His pairs are stronger than that of the deprotonated analogues. In addition, we agree with that the thermostability of ferritin crystal is the result integrative effect of multiple interaction such as electrostatic interaction and π stacking interaction. During the pH approach to 4.0, the electrostatic repulsive between adjacent ferritin may decrease as the pH are approaching to the pI (4.5) of ferritin, leading to the increased integrative attraction between two

ferritin. As compare to the increased His-His interaction, it's hard to specify their contribution by calculations experiment. Therefore, we re-write this part logically, and deem that the enhanced His-His interaction may associated to their thermostability but not the only decisive factor, it has been clarified in the revised manuscript.

3) In my previous comment 3, I stated: “The metal-coordination interactions mediated by Glu residues appear to be an afterthought rather than design. In fact, a reasonable first expectation would have been that the His residues should coordinate Ni and Cu before the Glu residues did (the authors really did not consider this initially?). Also, it appears that ionic strength itself is strong enough to drive 3D assembly. A good control experiment would be to eliminate the Glu's and examine the self-assembly in that case.” Although the authors state that “suggestion was followed”, it is not clear to me how they were followed.

First, the authors did not address my question as to whether they have really not considered that His residues would coordinate Ni and Cu before they were actually discovered to be involved in pi-pi interactions (this is particularly significant as the authors just recently published a paper on metal-mediated ferritin self-assembly. The current way the research is presented, I and many I and many readers will actually wonder on what basis the authors would have been able to predict that His-His interactions would dominate over His-Ni or His-Cu coordination. If it was a serendipitous discovery, the authors should say so and build their story from there (nothing is wrong with this). If it was not, the authors should provide some meaningful reason or discussion as to why they expected His-His stacking over His-metal coordination.

Second, I could not locate in the text whether the authors carried out the control experiments with eliminated Glu residues – if not, the authors explain why they didn't think that this was a necessary experiment. I believe this is actually critical as it relates directly to the authors' arguments regarding 2D vs 3D ferritin assemblies.

Response: We totally agree with your idea that nickel ions could coordinate with His. However, in this study, since His-His interactions could be adjusted by pH due to the unique pK_a of His residue, we plan to use His-His interactions to mediate the assembly of ferritin based on our previous study (On-Axis alignment of protein nanocage assemblies from 2D to 3D through the aromatic stacking interactions of amino acid residues. ACS Nano 2018, 12, 11323-11332) that π - π interactions such

as Phe-Phe and Tyr-Tyr can drive ferritin protein nanocages to form high-ordered arrays.

According to your suggestion, we made a new mutant named ^{T158H/E95A}MjFer in the revised manuscript, and found that no protein arrays was observed in solution under the same experimental conditions as ^{T158H}MjFer, demonstrating that Glu95 is a key amino acid residue responsible for coordination with nickel ions. We have added these results in Supplementary Figure 4.

4) Regarding the last point, the authors again respond to my original comment 4 as “Suggestion was followed in the revised version, and corrections have been made. We also gave an explanation of why protein assembly are so different between dilute solution and crystals.” They again don’t explain in the rebuttal letter what the explanation is. Importantly, the arguments they make about salt concentrations is more consistent with the importance of long-range electrostatic effects in governing ferritin self-assembly. Further, I don’t find the explanations regarding supersaturation compelling.

Response: Suggestion was followed in the revised version, and corrections have been made. The joint properties of protein and its solution environment, rather than properties of the protein molecule alone, responsible for the formation of crystal. Crystals form because of interactions between molecules and effected by pH, temperature, the concentration of salt and other solution components. We agree with your opinion that salt concentration is important to control self-assembly by long-range electrostatic effects. According to Durbin, S. D. and Feher, G. Protein crystallization. *Annu. Rev. Phys. Chem.* **47**, 171-204 (1996), salts play a complex role and can screen charges, enhance hydrophobic effects and lower the dielectric constant leading to the increase of long-range electrostatic effects, which all contribute to the formation of a crystal.

Besides, superficially, it seems that very strong and specific interactions would be desirable for crystal contacts. But the existence of several interactions would probably lead to formation of a gel rather than a crystal: large aggregates would form rapidly in solution and become entangled, inhibited by the strength of their bonds from reorganizing. According to Feher G, Kam Z. et al. Nucleation and growth of protein crystals: General principles and assays. *Method Enzymol.* **114**, 78-112 (1985), there is a dynamic equilibrium among aggregates of different size and structure in any solution of macromolecules. Crystals can form only if the solution is supersaturated, that is, if the supersaturation c/s exceeds 1, where c is the protein concentration and s the solubility (concentration

at equilibrium). Under normal crystallizing conditions, the free energy ΔG of a pre-crystalline aggregate first increases with size, then reaches a maximum, and thereafter decreases. This happens because in small aggregates, few intermolecular bonds at the surface could not make up for the decrease of entropy leading to the tendency to dissociate. And the larger ones tend to grow to nucleate because a smaller surface-to-volume ratio result in a higher proportion of intermolecular bonds. So the supersaturation is crucial to formation of a nucleus.

Response to comments from reviewer 3:

The authors have mostly answered my concerns. However, the reply to comment 3) is not sufficient.

“2D arrays in Figure 2b, should be verified with cryo-TEM or solution SAXS. The assembly could easily be a surface drying effect. Response: To confirm the TEM observation in Figure 2b, we used dynamic light scattering (Supplementary Figure 2) to determine protein assembly in solution, which is consistent with the TEM result. This finding excluded the possibility that the TEM observation is due to a surface drying effect.”

DLS data is not able to rule out ordered structures while drying. Cryo-TEM or solution SAXS data must be provided.

Response: Thanks for your suggestion. Your suggestion was followed, and we used both cryo-TEM and solution SAXS to characterize the 2D protein arrays observed by TEM, but unfortunately, we failed to observe the 2D protein arrays induced by only His-His interactions in solution. Therefore, we removed those results related to 2D protein arrays in the revised manuscript. However, our results demonstrated that the construction of the 3D protein frameworks is stemmed from a combination of the designed His-His interactions and serendipitous metal coordination.

REVIEWERS' COMMENTS:

Reviewer #2 (Remarks to the Author):

The authors seem to have made a good faith effort to revise their MS, which is considerably improved compared to the previous versions. Their further investigation of the 2D assemblies (in response to Reviewer 3's comments) showed that they were an artifact, which they have removed from the MS. They have also conducted additional mutagenesis studies to demonstrate the importance of Glu residues in Ni coordination. As I have stated before, I believe that the pH-tunable association of His residues is an interesting way to control protein self-assembly and represents a meaningful addition to literature in this area.

Yet, I still think that the scholarship in this paper (i.e., how it is written and how it puts existing literature into context) needs to be improved, but I will not make this a huge roadblock at this point, hoping that the authors keep this in mind in their future studies and manuscripts. However, I will still make a couple of points that the authors need to address to correct some misconceptions in the paper:

In the Conclusions, the authors state: "While a variety of different dimensional protein assemblies have been reported by using different noncovalent and covalent interactions as driving forces[21-33], how to construct binary protein and metal MOFs remains challenged. Here, we found an effective way to build protein-metal MOFs by introducing extra metal coordination bonds as the second driving force in conjugation with the designed noncovalent His-His interactions at the interactional interfaces."

First, there is an improper usage of the term MOF here. MOFs are porous frameworks that are composed of metal-based nodes and organic linkers. A protein-MOF consists of proteins, metal connectors and organic linkers (see for example Sontz et al., JACS 2015) as three separate components. What is constructed in this paper is not a MOF or a protein MOF as it does not have organic linkers – it is a 3D protein lattice that self-assembles via metal and non-covalent interactions, of which there are many different examples (natural or designed) – in fact, many protein crystals in the PDB databank self-assemble this way (i.e., through metal coordination and non-covalent interactions) - this of course does not make them MOFs. The authors also termed another metal-cluster-linked protein crystals as protein-MOFs in a previous publication. The usage of the term MOF is wrong in that case as well, because again there was no organic linker component. The authors should eliminate this wrong terminology in the current manuscript and not use it improperly in their future publications.

Second, I do not understand the statement "how to construct binary protein and metal MOFs remains challenged". What is meant by binary? If the authors refer to the use of metal ions in the design of self-assembled protein structures as being challenging (with or without additional non-covalent interactions), this statement is simply not correct as there are many prominent examples (which the authors do not cite) that speak to the contrary. The authors should perhaps state that there have been considerable advances in using metal coordination in controlling protein assembly and rendering it externally tunable (for example, Salgado et al. Acc. Chem. Res. 43, 661-672 (2010), Churchfield et al. Acc. Chem. Res. 52, 345-355 (2019), Brodin et al. Nat. Chem. 4, 375-382 (2012), Suzuki et al., Nature, 533, 369-373 (2016)), and that the current work adds to such examples of tunable protein self-assembly by exploiting pH-controllable His-His interactions.

The actual demonstration of such pH-controllability has been diminished by the finding that the 2D assemblies are likely artifacts, but this would still be an acceptable statement.

Reviewer #3 (Remarks to the Author):

It seems that author's were unable to provide conclusive experimental evidence to support their original hypothesis on the 2D sheets and have omitted the entire figure. This has led the authors to carry out rather major revision to the logic of the manuscript and the experimental data presented. For example:

Revision B: "Herein, by single His mutation on ferritin outer surface close to its C4 symmetry axes as shown in Figure 1, we implemented the His-His interactions within two neighboring protein molecules could be fine-tuned by pH in crystals (Figure 1b). and later it is stated "the designed His-His interactions hardly facilitate T158HMjFer protein nanocages to form high-ordered protein arrays in solution"

Revision A: "Herein, we implemented the construction of 2D protein arrays with ferritin nanocages as building blocks in solution by single His mutation on the protein outer surface close to its C4symmetry axes as shown in Figure 1.

This is significant change concerning the entire assembly mechanism (and not very well communicated). Furthermore, the authors still refer to the conversion from 2D lattices in figure 4 caption, which cannot be correct.

When 2D lattice conversion is omitted from the manuscript it reduces to a report that demonstrates how the added His residue affects the crystallization of MjFer.

Response to comments from reviewer 2:

Yet, I still think that the scholarship in this paper (i.e., how it is written and how it puts existing literature into context) needs to be improved, but I will not make this a huge roadblock at this point, hoping that the authors keep this in mind in their future studies and manuscripts.

Response: Thanks for your suggestion, and we have revised the section of Introduction in the revised manuscript.

However, I will still make a couple of points that the authors need to address to correct some misconceptions in the paper:

In the Conclusions, the authors state: “While a variety of different dimensional protein assemblies have been reported by using different noncovalent and covalent interactions as driving forces[21-33], how to construct binary protein and metal MOFs remains challenged. Here, we found an effective way to build protein-metal MOFs by introducing extra metal coordination bonds as the second driving force in conjugation with the designed noncovalent His-His interactions at the interactional interfaces.”

First, there is an improper usage of the term MOF here. MOFs are porous frameworks that are composed of metal-based nodes and organic linkers. A protein-MOF consists of proteins, metal connectors and organic linkers (see for example Sontz et al., JACS 2015) as three separate components. What is constructed in this paper is not a MOF or a protein MOF as it does not have organic linkers – it is a 3D protein lattice that self-assembles via metal and non-covalent interactions, of which there are many different examples (natural or designed) – in fact, many protein crystals in the PDB databank self-assemble this way (i.e., through metal coordination and non-covalent interactions) - this of course does not make them MOFs. The authors also termed another metal-cluster-linked protein crystals as protein-MOFs in a previous publication. The usage of the term MOF is wrong in that case as well, because again there was no organic linker component. The authors should eliminate this wrong terminology in the current manuscript and not use it improperly in their future publications.

Second, I do not understand the statement “how to construct binary protein and metal MOFs remains challenged”. What is meant by binary? If the authors refer to the use of metal ions in the design of self-assembled protein structures as being challenging (with or without additional

non-covalent interactions), this statement is simply not correct as there are many prominent examples (which the authors do not cite) that speak to the contrary. The authors should perhaps state that there have been considerable advances in using metal coordination in controlling protein assembly and rendering it externally tunable (for example, Salgado et al. *Acc. Chem. Res.* 43, 661-672 (2010), Churchfield et al. *Acc. Chem. Res.* 52, 345-355 (2019), Brodin et al. *Nat. Chem.* 4, 375-382 (2012), Suzuki et al., *Nature*, 533, 369–373 (2016)), and that the current work adds to such examples of tunable protein self-assembly by exploiting pH-controllable His-His interactions

Response: Thanks for your suggestions. All of your suggestions have been followed. First, we have removed term of MOFs in the revised manuscript. Second, the significance of constructing assemblies has been added to the revised manuscript.

Response to comments from reviewer 3:

It seems that author's were unable to provide conclusive experimental evidence to support their original hypothesis on the 2D sheets and have omitted the entire figure. This has led the authors to carry out rather major revision to the logic of the manuscript and the experimental data presented. For example:

Revision B: "Herein, by single His mutation on ferritin outer surface close to its C4 symmetry axes as shown in Figure 1, we implemented the His-His interactions within two neighboring protein molecules could be fine-tuned by pH in crystals (Figure 1b). and later it is stated "the designed His-His interactions hardly facilitate T158HMjFer protein nanocages to form high-ordered protein arrays in solution"

Revision A: "Herein, we implemented the construction of 2D protein arrays with ferritin nanocages as building blocks in solution by single His mutation on the protein outer surface close to its C4symmetry axes as shown in Figure 1.

This is significant change concerning the entire assembly mechanism (and not very well communicated). Furthermore, the authors still refer to the conversion from 2D lattices in figure 4 caption, which cannot be correct.

When 2D lattice conversion is omitted from the manuscript it reduces to a report that demonstrates how the added His residue affects the crystallization of MjFer

Response: Thanks for your suggestions. In revision B, we verify that His-His interactions is too weak to facilitate protein assemblies in solution, but the specific His-His interactions can be clearly observed in crystals. Protein-protein interactions in crystals can be fine-tuned by pH as demonstrated by X-ray protein crystal structures at pH 4.0, 7.0, and 9.0. Such difference in protein-protein interactions between crystals and solution is most likely derived from their completely different experimental conditions. We have added this interpretation in the revised version. In crystals, a significant change in stacking pattern between protein molecules is the precise reflection of how the designed His-His interactions perform as a function of pH. We have revised the manuscript and corrected caption in Figure 4.